# Application of Covalent Organic Frameworks (COFs) as Dyes and Additives for Dye-Sensitized Solar Cells (DSSCs)

**DOI:** 10.3390/nano13071204

**Published:** 2023-03-28

**Authors:** Diogo Inácio, Ana Lucia Pinto, Ana B. Paninho, Luis C. Branco, Sunny K. S. Freitas, Hugo Cruz

**Affiliations:** 1LAQV-REQUIMTE, Department of Chemistry, NOVA School of Science and Technology, NOVA University of Lisbon, Campus de Caparica, 2829-516 Caparica, Portugal; 2Instituto de Química—UFRJ Avenida Athos da Silveira Ramos, n° 149, Bloco A—7° Andar Centro de Tecnologia—Cidade Universitária, Ilha do Fundão, Rio de Janeiro 21941-909, Brazil

**Keywords:** dye, additive, photoanode, COVALENT Organic Frameworks (COFs), dye-sensitized solar cell (DSSC)

## Abstract

Five Covalent Organic Frameworks (COFs) were synthesized and applied to Dye-Sensitized Solar Cells (DSSCs) as dyes and additives. These porous nanomaterials are based on cheap, abundant commercially available ionic dyes (thionin acetate RIO-43, Bismarck brown Y RIO-55 and pararosaniline hydrochloride RIO-70), and antibiotics (dapsone RIO-60) are used as building blocks. The reticular innovative organic framework RIO-60 is the most promising dye for DSSCs. It possesses a short-circuit current density (J_sc_) of 1.00 mA/cm^2^, an open-circuit voltage (V_oc_) of 329 mV, a fill factor (FF) of 0.59, and a cell efficiency (η) of 0.19%. These values are higher than those previously reported for COFs in similar devices. This first approach using the RIO family provides a good perspective on its application in DSSCs as a dye or photoanode dye enhancer, helping to increase the cell’s lifespan.

## 1. Introduction

In recent years, porous materials have played important roles in several relevant fields in both academia and industry. Depending on the demand, different kinds of materials have been designed and applied in order to meet sustainability needs [1].

A special and more robust class of porous organic materials, Covalent Organic Frameworks (COFs), has shown great versatility due to having a high chemical stability, large surface area, ordered porosity, and diverse structures [2,3,4,5,6]. These materials have emerged as a new class of nanostructured porous materials that are capable of forming highly porous extended networks through covalent bonds between light elements. Their pre-established design, facile synthesis, and range of available building-blocks make them good candidates for use for diverse purposes, such as in technologies related to the environment and energy [7,8,9,10,11,12,13]. These materials usually present a two-dimensional (2D) or three-dimensional (3D) porous crystalline structure [4,5,6].

COFs and their derivatives (Covalent Organic Networks [14]/Nanosheets, Porous Organic Polymers, among others) can be subdivided into ionic or nonionic systems [4,15,16]. Additionally, CON nanomaterials also demonstrate extraordinary porosity, which can be tuned and maintained by their excellent chemical stability [4,5,6,17]. In particular, their 2D organization with conductive properties gives them countless prospective applications and fundamental novelties for use in different electronics and sensors. Furthermore, such materials can be electroactive even if they do not possess donating ions through the framework [18,19]. In general, heteroatoms, such as N, O, S and others, cause crystal channel direction with a high electron density, thus accelerating the ion mobility rate through the porous channel. These ionic materials have rarely been reported, even though they have great potential for use in innovative applications, such as sensing, solid-state electrolytes, and selective gas capture, among others.

The right choice of the building blocks and their functional groups is crucial for the good performance of these materials in suitable applications. In this context, lamellar COFs (2D) based on amine and aldehyde groups were selected for this work. Based to previous experience, porous materials known as RIOs (Reticular Innovative Organic compounds) were synthesized from ionic commercial dyes (thionin acetate, Bismarck brown Y and pararosaniline hydrochloride), and an electroactive commercial antibiotic (dapsone) was used as the amine linker. Dapsone is a sulfone-based drug that is used for the treatment of Hansen’s disease (leprosy), among other skin treatments [20]. The electroactive 2,6-anthraquinone has been widely used for energy purposes [21]. On the other hand, the highly stable triformylphloroglucinol was selected as the aldehyde linker due to its irreversibility, making this framework more robust [4,6].

There are donor-acceptor-type COFs (D-A COFs), in which the subunits are an acceptor and an electron donor [22], in addition to other types of charge-carrying structures, such as Ionic Covalent Organic Frameworks (ICOFs), which possess an ionic component in their structures [23].

It is important to note that lamellar COFs (2D) have a large number of double bonds, allowing π-stacking interactions and facilitating electron transport as well as absorption into the visible region of the light spectrum. Taking advantage of these properties, the aim of this work is related to the design and synthesis of photo and electroactive COFs for application as alternative dyes or electrolyte additives in dye-sensitized solar cells.

Solar cells can be divided into four generations according to their different structural and functional characteristics:

(1)The *first generation* is made up of the most commonly used commercial silicone-based materials, which are efficient but have a high cost;(2)The *second generation* comprises very thin and flexible cells with a lower efficiency than the previous one while being less expensive [24];(3)*Third-generation* cells are based on intermolecular interactions and have a considerably lower production cost compared to the first-generation cells [25,26,27];(4)The *fourth generation* is a more recently developed class of cells that use the interaction between organic and inorganic materials to obtain a greater efficiency and stability [28].

DSSCs belong to the third generation, which differs from the first two due to the possibility of adjusting the cell’s absorption band as well as the possibility of producing flexible cells. In addition, the complex set of intermolecular interactions between the various components of the cell structure permits fine-tuning, resulting in more precise control over their performance and efficiency [25].

In 1991, Brian O’Regan and Michael Grätzel first proposed the idea of a solar cell based on a mesoporous titanium dioxide (TiO_2_) thin film covered by dye molecules. These cells are low-cost and highly efficient compared to other more expensive photovoltaic devices, achieving photon-to-electrical conversions of over 80% and solar-to-electrical efficiencies of between 7.1 and 7.9% [29]. In addition, DSSCs are easy to manufacture and scale up, which, as expected, has captured significant attention from the scientific community. To date, the cell efficiency record is around 14% [30]. Following the pioneer work of Grätzel, a basic understanding of the working principles of DSSCs was gained using several types of dye, such as original ruthenium dyes [31,32], porphyrins [33,34,35], anthocyanins [35,36,37,38,39,40], pyranoanthocyanin [40,41,42], and polyoxometaleytes [43], among others. Additionally, various carbon forms, carbon nanotubes, graphite, graphene, graphene doped with oligothiophene [44], and materials such as 2D-Nanolayer (2D-NL)-Based Hybrid Materials (either electron-transport layers (ETLs) or hole-transport layers (HTLs)), can also be used in dye-sensitized solar cells (DSSCs). However, they present some drawbacks, such as low stability and a low photo conversion efficiency (PCE), because the stability of these materials is one of the major challenges following the development of DSSCs [45].

Another important factor is the correct choice of the electrolyte to achieve highly efficient DSSCs.

In parallel, the selection of the redox mediator has been reported as a crucial parameter. Examples of redox mediators include cobalt (II/III) [46,47] or copper (I/II) [48,49] complexes. These could replace the most common I^−^/I_3−_, gel-based quasi-solid-state electrolytes [50,51,52,53] or even hole transport materials [54,55]. Despite this, the most efficient DSSC uses a liquid electrolyte [56,57].

In this work, COFs were integrated into DSSCs with the following structure: (a) a conductive FTO (Fluorine-doped Tin Oxide) glass electrode, upon which was deposited (b) an anatase-based blocking layer, (c) a 15 μm layer of titanium dioxide nanoparticles, (d) dye (to capture energy from sunlight), (e) an electrolyte (an iodine–iodide solution), and (f) a platinum counterelectrode, deposited on (g) another FTO glass. With this structure in mind, the intention was to replace the dye component with different COFs and understand their impacts on the overall performance of the DSSC (see Figure 1).

To date, the implementation of COFs in DSSCs has rarely been explored. Herein, different 2D COFs named RIO-43, RIO-55, RIO-70, RIO-60, and DAAQ-COF were tested as dyes, thus exploring their versatility for use in DSSC applications (see Figure 2).

## 2. Materials and Methods

### 2.1. DSSC Fabrication and Photovoltaic Characterization

The conductive FTO-glass (TEC7, Greatcell Solar) used for the preparation of the transparent electrodes was cleaned by an ultrasound process in three steps: (1) immersion in soapy water for 15 min; (2) immersion in Milli-Q water for 15 min; and finally, (3) immersion in ethanol for 5 min. To prepare the anodes, the conductive glass plates (area: 15 cm × 4 cm) were immersed in a TiCl_4_/water solution (40 mM) at 70 °C for 30 min, washed with water and ethanol, and sintered at 500 °C for 30 min. Afterwards, the TiO_2_ nanocrystalline layers were deposited on these pretreated FTO plates by screenprinting the transparent titania paste (18NR-T, Greatcell Solar) using a frame with polyester fibers with 43.80 mesh per cm^2^. The TiO_2_-coated plates were gradually heated up to 325 °C. Then, the temperature was increased to 375 °C in 5 min, and afterwards, it was increased to 500 °C. The plates were sintered at this temperature for 30 min and finally cooled down to room temperature, Figure 3.

The dye was adsorbed to the TiO_2_ film by soaking in a 0.5 mM dye solution for the required time, Figure 4.

Each counterelectrode consisted of an FTO-glass plate (area: 2 cm × 2 cm) in which a hole (1.5 mm diameter) was drilled. The perforated substrates were washed and cleaned with water and ethanol in order to remove any residual glass powder and organic contaminants. The transparent Pt catalyst (PT1, Greatcell Solar) was deposited on the conductive face of the FTO-glass by a doctor blade: one edge of the glass plate was covered with a strip of adhesive tape (3 M Magic) to both control the thickness of the film and mask an electric contact strip. The Pt paste was spread uniformly on the substrate by sliding a glass rod along the tape spacer. The adhesive tape strip was removed, and the glasses were heated at 550 °C for 30 min, Figure 3.

After this step, the anode was ready for cell assembly.

The photoanode and the Pt counterelectrode were assembled into a sandwich type arrangement (Figure 4) and sealed (using a thermopress) with a hot melt gasket made from Surlyn ionomer (Meltonix 1170-25, Solaronix SA), Figure 4. The electrolyte was prepared by dissolving the redox couple, I^−^/I_2_ (0.8 M LiI and 0.05 M I_2_), in an acetonitrile/valeronitrile (85:15, % *v*/*v*) mixture. The electrolyte was introduced into the cell via backfilling under vacuum through the hole drilled into the back of the cathode. Finally, the hole was sealed with adhesive tape.

### 2.2. Optimization of COF Adhesion to the Photoanode

The prepared COFs (RIO-55, RIO-60 and COF DAAQ-TFP) were selected for adhesion to the photoanode.

For the three reactions, the required amount was approximately the same: 0.2 mmol of amine and 0.3 mmol of TFPG as well as the volume of solvent (8 mL of 1,4-Dioxane and 2 mL of 6 M acetic acid). The optimized experimental protocol was followed: weigh out the necessary amounts of amine and TFFG, add both inside the reactor, and then add the solvents in the following order: 1,4-dioxane and then 6 M acetic acid. After this step, the reactor was closed and left under stirring and heating at 120 °C for 3 h. After this time period, the reactor was opened, the magnetic stirrer was removed, and the anodes were added. The reactor was closed and left in the bath without stirring for 5 h. After this time, the anodes are removed, washed with ethanol, and left to dry in a dark environment. Then, the glasses were mounted in the cells and tested with the COFs as dye substitutes. The materials resulting from these syntheses were individually filtered and dried in the same way as the previous ones.

For optimization of the process, some parameters were considered, such as the precrystallization period of the COF before the addition of the anode (30 min, 2, and 24 h) and the duration that the anode remains inside the reactor (the deposition time on the anode varied between 30 min, 5, and 24 h).

RIO-60 was selected as the model compound for the optimization test, and selected amounts of the precursors (0.02 mmol of dapsone and 0.03 mmol of TFFG for 8 mL of 1,4-dioxane plus 2 mL of acid acetic 6M) were considered.

### 2.3. N719 Deposition on the Photoanode

The N719 dye was selected as a reference for comparison with the prepared materials. In addition, the starting materials used for the three most promising COFs were deposited.

The photoanode was soaked in the dye solution for the desired period of time (similar to the COF deposition time on the photoanode).

### 2.4. Use of COF as an Additive (Interface between Dye and Photoanode)

Finally, the three syntheses with the best results from Section 2.2 with RIO-60 were repeated. However, before the assembly of the DSSC, the COF-impregnated photoanode was immersed in N719 dye for 2 h, see Figure 5. Thus, photoanodes were formed with COF bound to TiO_2_ and with impregnated dye, see Figure 5. These anodes were dried in an oven at 120 °C for 20 min. Similar conditions were used for all assembled and tested DSSCs.

### 2.5. Photoelectrochemical Measurements

Current–voltage curves were recorded with a digital Keithley SourceMeter multimeter (PVIV-1A) (Newport) connected to a PC. Simulated sunlight irradiation was provided by an Oriel solar simulator (Model LCS-100 Small Area Sol1A, 300 W Xe Arc lamp equipped with AM 1.5G filter, 100 mW/cm^2^) (Newport). Cell testing was performed in a closed electrical circuit and accurately placed under the light flux to ensure correct and efficient measurement of the cell.

### 2.6. Chemicals

All commercially available reactants were of high purity, used without further purification, and stored under inert conditions. The synthesis of RIO-43 and RIO-55 [4], RIO-70 [6], and COF DAAQ [21] was performed according to previous literature.

### 2.7. Synthesis of Triformylphloroglucinol (TFPG)

This synthesis was performed as follows: 60 mL of trifluoroacetic acid (TFA) was added to a round-bottom flask (500 mL) under an ice bath. Then, 9.26 g of hexamethylenetetramine (HMTA) was slowly added under stirring until total solubilization of the amine was achieved. After this, 3.85 g of phloroglucinol was added. The mixture was placed in a silicone bath at 100 °C, where it remained under reflux for 16 h. After this period, 95 mL of HCl 3M was added, and the final solution was orange in color. After 4 h, the reaction was stopped, and the purification steps were followed. To the solution, 200 mL of DCM (dichloromethane) and ~500 mL of distilled water were added in order to wash and separate the organic phase. This washing step was repeated at least 3 times. With a separation funnel, the organic phase was collected, and the aqueous phase was continuously washed. The organic phase was a yellow color, and after drying the DCM using a rotary evaporator, a pale-yellow solid was obtained (~60%). The solid product was left under vacuum for ~8 h. Appendix A presents the results of the characterization by diffuse reflectance (left) and FTIR for TFPG.

### 2.8. Synthesis of RIO-60

To the best of the authors’ knowledge, this compound has not, to date, been reported in the literature. As such, its synthesis and characterization are described in detail. The reaction schematics are presented in Figure 6. The synthetic procedure was similar to that used for other RIOs: 40 mg (0.16 mmol) of Dapsone (97%, Sigma-Aldrich, St. Louis, MI, USA) and 50 mg (0.24 mmol) of synthesized TFPG were added to a reactor (high pressure vessel, ChemGlass, 48 mL), followed by the addition of the solvent (8 mL of 1,4-Dioxane; 99.5% Merck) and, finally, the catalyst (2 mL of acetic acid 6M; 99.7% Honeywell). The reactor was sealed and remained under heating (at 120 °C) and stirring for 10 min. After this time, the reactor was opened, the magnetic stirrer was removed, and a TiO_2_-based anode (blocking layer and TiO_2_) was inserted. Then, the vessel was closed, and the reaction proceeded at 120 °C without stirring for 20 h.

After the predicted reaction time, the glass was removed and washed with ethanol. Then, it was left to dry in an environment protected from light and used to assemble a cell where RIO-60 was used as a dye substitute. This material was an intense yellow color (see Figure 6).

Regarding the solid resulting from the reaction, the COF (bulk) was filtered and then packed in filter paper and washed with ethanol using a Soxhlet for 8 h. When the material was visually without starting materials, it underwent supercritical drying with CO_2_ and maintained its yellowish color.

This prepared COF (RIO-60) was characterized by different spectroscopic techniques in order to elucidate its chemical structure, as can be observed in Figure 7.

Figure 7 presents the physical-chemical characterization of RIO-60 by (a) FTIR, (b) solid state ^13^C NMR, (c) PXRD, and (d) the BET method for the surface area N_2_ adsorption–desorption isotherm and pore size distribution. Figure 7a presents FTIR spectroscopy characterization, which is frequently used to determine the compositional structure of materials. The spectrum obtained in Figure 7a for RIO-60 shows the characteristic signal for RIO-60. The broad band at 3480 cm^−1^ is attributed to -C-N, while the signal at 1628 cm^−1^ is related to -C=O. The bands at 1330 and 1180 cm^−1^ are associated with the -SO_2_ asymmetric and -SO_2_ symmetric stretch. Figure 7b presents the solid-state ^13^C CP-MAS NMR spectrum characterization of RIO-60, which is also frequently used to complete the determination of the compositional structure of COFs. The spectrum obtained for RIO-60 carbon is presented in Figure 7b. Different colors associated with carbon represent the following: pink dots, 184.88 ppm; green dots, 142.57 ppm; blue dots, 138.04 ppm; red dots, 130.03 ppm; purple dots, 119.63 ppm; and yellow dots, 108.08 ppm.

Characterization by PXRD using powder XRD is shown in Figure 7c. The diffractogram acquired shows a low crystallinity profile with a broad peak at approximately 25°. Figure 7d presents characterization by the Brunauer–Emmet–Teller (BET)-specific surface area obtained by the N_2_ adsorption–desorption isotherm at 77 K RIO-60. The pore size distribution (see the inset of Figure 7d), calculated by NLDFT model from the N_2_ desorption branch at 77 K on carbon cylindrical/slit pores, showed a pore size of 13 Å. Afterwards, the COFs were characterized, and to ensure that the maximum pore diameter was always reached, it was dried with supercritical CO_2_.

### 2.9. Supercritical Drying of the Synthesized COF with CO_2_

Supercritical fluid drying of the COFs was carried out using a semicontinuous high-pressure apparatus that is described in detail elsewhere [58]. Briefly, the high-pressure apparatus was composed of a cylindrical extractor with a volume of 12 cm^3^ that was positioned vertically in a thermostated water bath and heated to the desired temperature by means of a controller that maintained the temperature within ±1 K (JP Selecta Termotronic). The COF was placed at the bottom of the extractor, which was always immersed in 5 mL of ethanol in order to avoid the risk of solvent evaporation from the COF prior to coming into contact with the CO_2_ atmosphere. The addition of CO_2_ was conducted using a pneumatic compressor (Electrolux) connected to the reactor. The pressure in the system was measured with a pressure transducer (Setra Datum 2000 TM), which was calibrated at between 0 and 34.3 Mpa (with a precision of ±0.1% at the lowest pressure). The CO_2_ flow was controlled by two high-pressure valves, connected directly to the sealing system at the top of the extractor and further connected to a gas flow meter. The two high-pressure valves were slowly released and manipulated to control the CO_2_ flow and maintain a constant pressure inside the system, see Figure 8. The process was carried out for three hours with a constant CO_2_ flow rate of 3 g min^−1^. A temperature of 313.2 K and a pressure of 15 Mpa were selected to ensure the supercritical conditions of the binary system (ethanol + CO_2_). At the end of the process, the system was slowly depressurized, and the COF was removed from the extractor.

## 3. Results and Discussion

The standard DSSC assembly is based on a sandwich architecture type. Herein, we describe the fabrication of the devices using the following configuration: (1) anode with a nanocrystalline TiO_2_ film; (2) sensitizer N719 [Di-tetrabutylammonium cis-bis(isothiocyanato)bis(2,2′-bipyridyl-4,4′-dicarboxylato)ruthenium(II)], adsorbed to the TiO_2_; (3) I^-^/I_3_^-^ based electrolyte; and (4) a Pt counterelectrode. Firstly, the starting materials (BBY-Bismarck Brown Y, DAPSO-Dapsone, DAAQ-2,6-diaminoanthroquinone and TFPG) used for the synthesis of COFs (RIO-55, RIO-60 and COF-DAAQ) were tested. The commercial dye N719 was, at all times, used as a reference. The anodes were soaked in the dye solutions for 5 h (Table 1).

To evaluate the ability of the COFs (and their starting materials) to act as dyes for DSSCs, different parameters can be used, such as the fill factor (FF), efficiency (η, %), the open-circuit voltage (*V*_oc_, mV), and the short-circuit current density (J_sc_, mA/cm^2^) measured under AM 1.5G solar light (100 mW/cm^2^).

From Table 1 and Figure 9, it can be observed that TFPG showed the best photovoltaic performance. This can be attributed to the presence of aldehyde groups (hydroxide -OH and =O) which are generally good anchoring groups for titanium dioxide (TiO_2_). Furthermore, TFPG is a small organic molecule with the ability to make π-stacking interactions with a large π conjugation. This may have contributed to the higher verified J_sc_, since an increase in π conjugation can contribute to a high electron flux (see Table 1).

The second-best result was obtained for BBY, which is a dye that is commonly used for photochemical and photoelectrochemical applications. From all of the starting materials tested, DAPSO had the poorest performance.

Table 2 and Figure 10 present the results for the COFs used as dyes in DSSCs. Despite being successfully adsorbed onto the TiO_2_-anode, the results obtained for RIO-43 were not as good as expected. A possible explanation for this is that RIO-43 can behave electrostatically, which could lead to aggregation, reducing the electron flux and finally having a direct impact on the photovoltaic performance.

In the case of RIO-70, since it is a partial carbocation, only a part of the porous structure contains chloride as a counteranion. This can result in part of the reticular structure of the *framework* remaining partially charged, thus ultimately presenting a lower charge distribution. In contrast, the other COFs presented in Table 2 (RIO-55, RIO-60 and COF-DAAQ presented in Figure 5), which contain oxygen and sulfur in their structures, showed better photovoltaic performances. This can be attributed to higher stabilization of the DSSC due to the interaction with the electrolyte/electrode interface and/or improved anchoring with the TiO_2_-based photoanode.

In general, RIO-60 showed the highest J_sc_ value, while RIO-55 and COF-DAAQ presented FF values close to that of the reference dye N719. COF-DAAQ showed a V_oc_ value similar to that of the reference N719 (Table 1).

For the best systems (RIO-55, RIO-60 and COF-DAAQ), an optimized deposition step was performed, resulting in a slight improvement in the results, as presented in Table 3 and Figure 11.

For RIO-60, the adsorption conditions were optimized as indicated in Table 4. Batches 1, 2, and 3 were related to RIO-60 with different deposition times (30 min, 2 h, and 24 h, respectively) prior to crystallization at the TiO_2_ surface.

The results of the deposition optimization procedure are presented in Table 5 and Figure 12. The results were verified for batches 1 and 2, and considering the increase in the deposition time for RIO-60, they seem to indicate increased passivation of the photoelectrode surface. This justifies the observed increase in the FF values, corresponding to a decrease in the recombination processes.

Based on the obtained results, it is possible to predict a correlation between the crystallization time and current density, where lower deposition times are preferential. This can be explained by the anchoring of the compounds to the TiO_2_ surface, where a monolayer is expected to form extremely swiftly with only a few points fixed onto the surface. The trialdehyde compound can occupy anchorage points, forming the first layer on TiO_2_, followed by interactions and bonds with the amine. Nevertheless, dapsone possesses oxygenated groups, which can also contribute to anchoring onto TiO_2_. However, the V_oc_ values seem to reach their highest points within a few hours of crystallization; as such, the average η is similar under these two conditions (B1 and B2).

RIO-60 was also used as an additive (an interface between the dye and TiO_2_). Some of the previous batches were mixed with N719 in order to evaluate the final photovoltaic performance, as indicated in Table 6 and Figure 13.

It is possible to conclude that, when using the reference dye N719 as a light harvester, the V_oc_ results do not increase with prior crystallization. After just 30 min of crystallization, the obtained V_OC_ results reached values very close to that obtained for the reference dye alone. The efficiency (η) and J_sc_ increased by more than two times for longer crystallization times. Concerning the COF deposition time, it significantly increased the efficiency and J_sc_ results, while the V_oc_ was maintained and the FF decreased.

## 4. Conclusions

The versatility and greater stability of these materials are particularly relevant for solar cells and similar applications. Herein, a first attempt at the implementation of RIOs in DSSCs was performed.

The prior crystallization of the COF derivatives for 24 h followed by absorption generated poor photovoltaic performances. The best methodology for the deposition of COF derivatives as dyes in a DSSC required slight prior crystallization followed by a half-hour deposition.

The most promising COF was RIO-60. Firstly, as a dye, it was shown to have a short-circuit current density (J_sc_) of 1.0 mA/cm^2^, a cell efficiency (η) of 0.19%, an open-circuit voltage (V_oc_) of 329 mV, and a fill factor (FF) of 0.59.

As a dye, RIO-60 presented a promising photovoltaic performance (0.19%), compared to previous results for TT-COF-PCMB (cell efficiency of 0.05%) [59]. These results emphasize the importance of using design task-specific RIOs in order to improve the solar cell application.

It is important to note that RIO-60 was also used as an interface between the reference dye N719 and titanium dioxide, and promising results were attained. The batch was reacted for 2 h, followed by annealing/crystallization to TiO_2_ for half an hour with an N719 adsorption time of 2 h presented the highest FF values. However, it also presented lower V_oc_, J_sc_, and η values when compared to the use of N719 alone. Future work will involve the use of novel porous nanomaterials starting from cheap and abundant building blocks in order to obtain a better solar cell performance, leading to a long-lasting solar cell.

## Figures and Tables

**Figure 1 nanomaterials-13-01204-f001:**
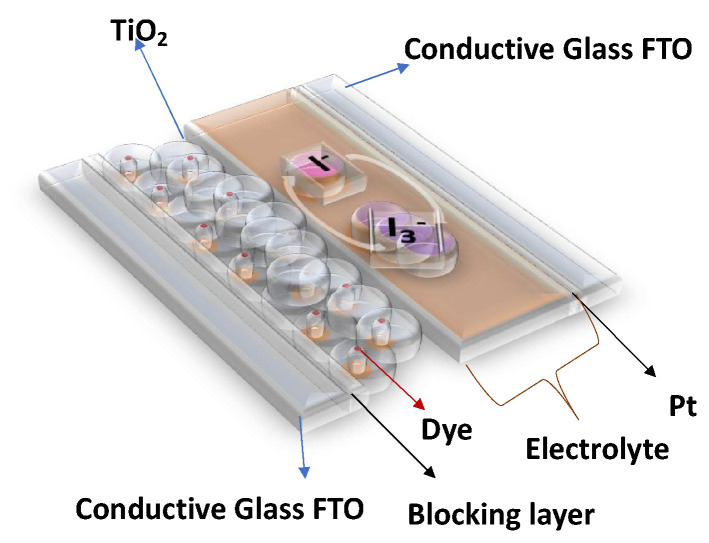
General structure of a Dye-Sensitized Solar Cell (DSSC).

**Figure 2 nanomaterials-13-01204-f002:**
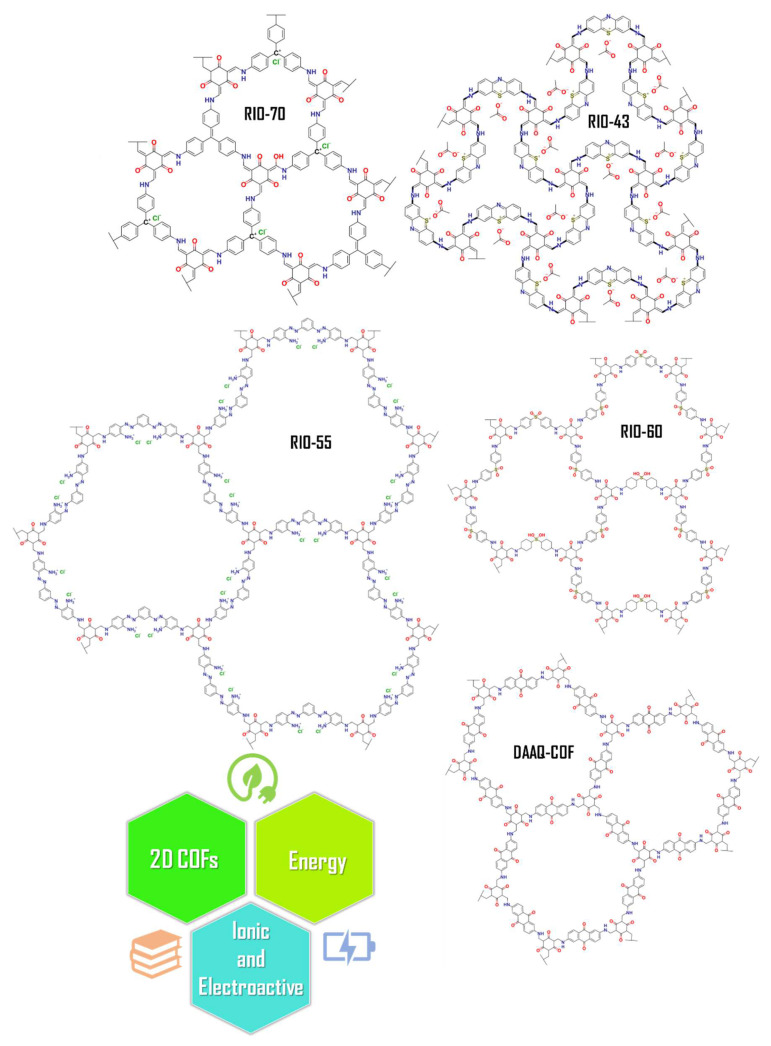
Chemical structures of selected 2D COFs with different pore sizes. The ionic and electroactive porous materials could be applied in the energy field, such as in solar cells, working on different interfaces.

**Figure 3 nanomaterials-13-01204-f003:**
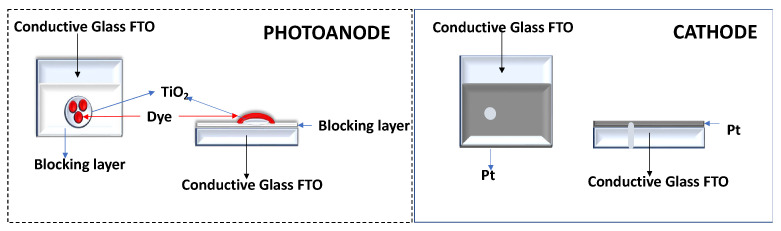
Representative schemes of the photoanode and cathode.

**Figure 4 nanomaterials-13-01204-f004:**
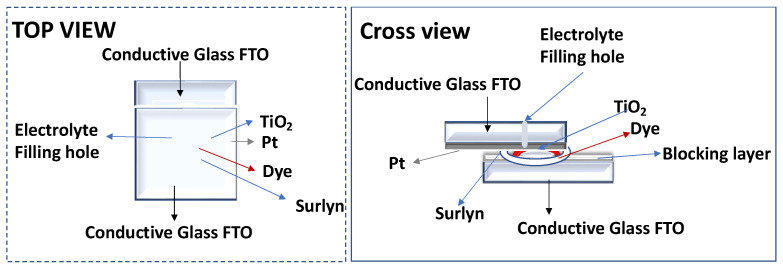
Schematic representation of DSSC assembly.

**Figure 5 nanomaterials-13-01204-f005:**
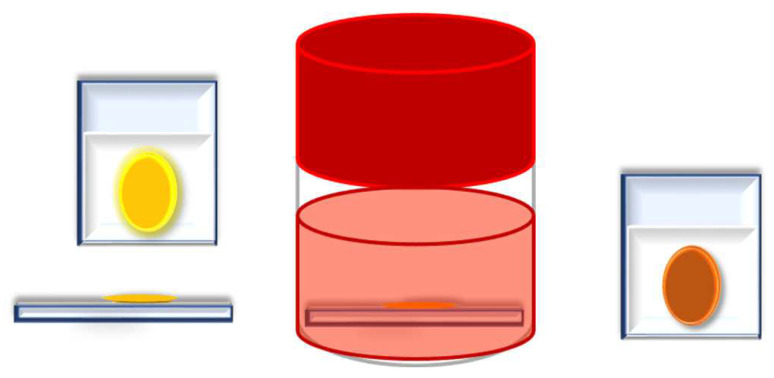
Impregnation of COF as an additive (interface between the dye and photoanode).

**Figure 6 nanomaterials-13-01204-f006:**
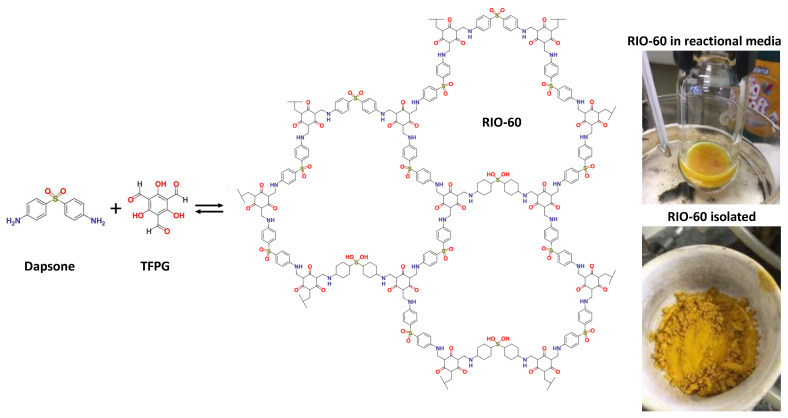
Synthesis of RIO-60. Dapsone is an angled building-block that forms more closed pores and has a structure that is susceptible to crystalline disorders. Additionally, it has interesting properties, such as photoactivity and catalysis.

**Figure 7 nanomaterials-13-01204-f007:**
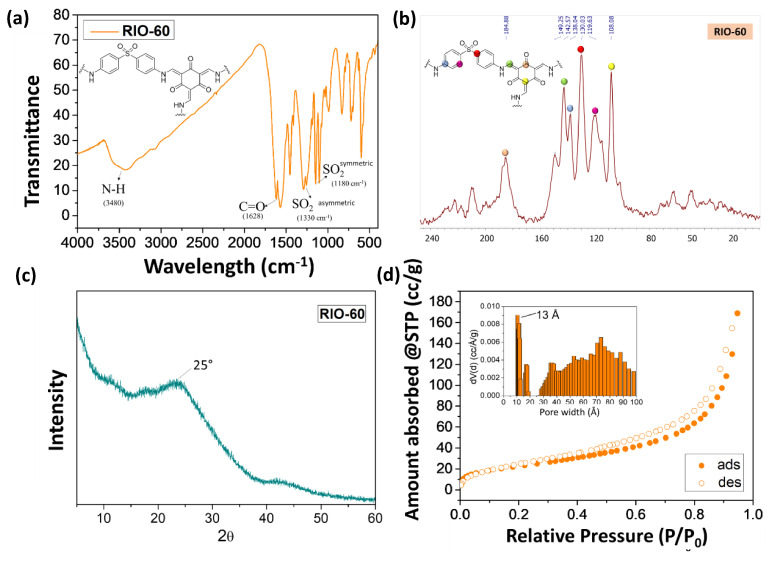
Characterization of RIO-60: (**a**) FTIR, (**b**) solid state ^13^C NMR, (**c**) PXRD, and (**d**) the BET method used for surface area N_2_ adsorption–desorption isotherm and pore size distribution.

**Figure 8 nanomaterials-13-01204-f008:**
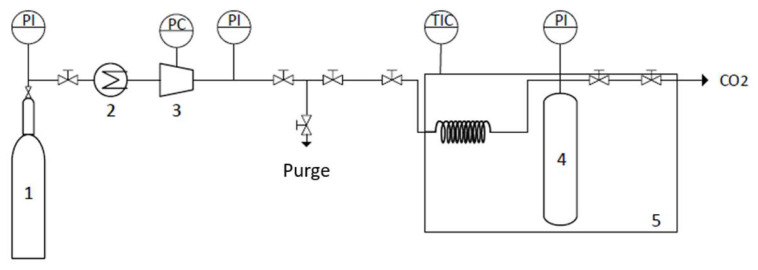
Schematic representation of the scCO_2_ drying apparatus: (1) CO_2_ tank, (2) Cooler, (3) Compressor, (4) Reactor, and (5) Thermostat water bath.

**Figure 9 nanomaterials-13-01204-f009:**
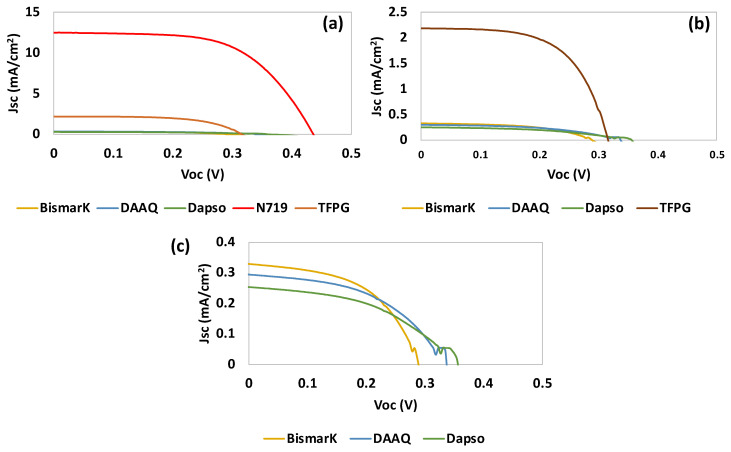
Photocurrent–voltage (J–V) curves of DSSCs based on (**a**) the starting materials and the use of N719 as dye after adsorption for 5 h, (**b**) the starting materials, (**c**) the starting materials without TFPG. Tests were conducted under simulated AM 1.5 G illumination at 100 mW/cm^2^.

**Figure 10 nanomaterials-13-01204-f010:**
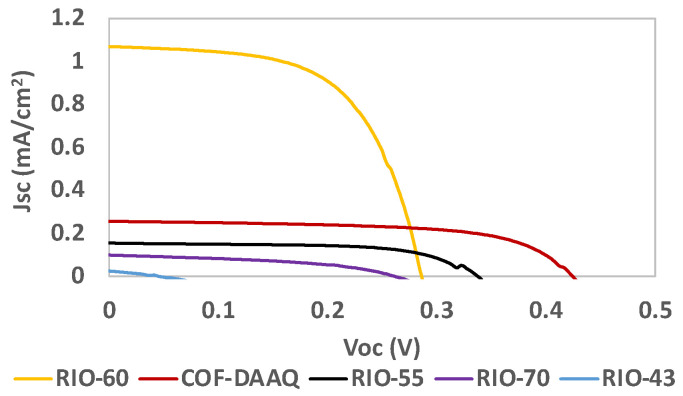
Photocurrent–voltage (J–V) curves of DSSCs based on COFs as the dye replacement under simulated AM 1.5 G illumination at 100 mW/cm^2^.

**Figure 11 nanomaterials-13-01204-f011:**
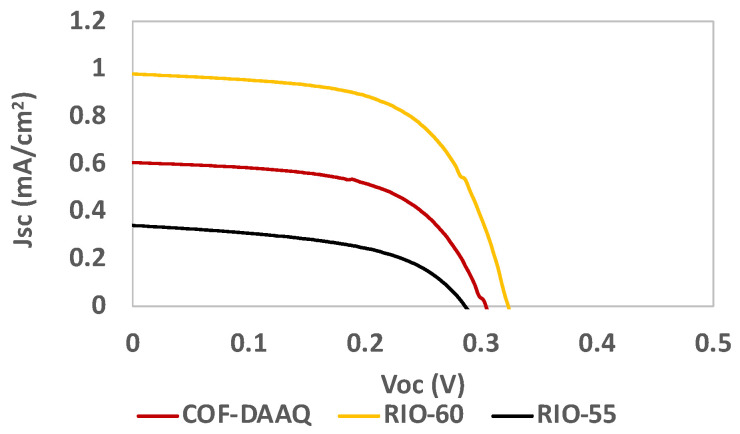
Photocurrent–voltage (J–V) curves of DSSCs based on COFs as the dye replacement using an optimized deposition step under simulated AM 1.5 G illumination at 100 mW/cm^2^.

**Figure 12 nanomaterials-13-01204-f012:**
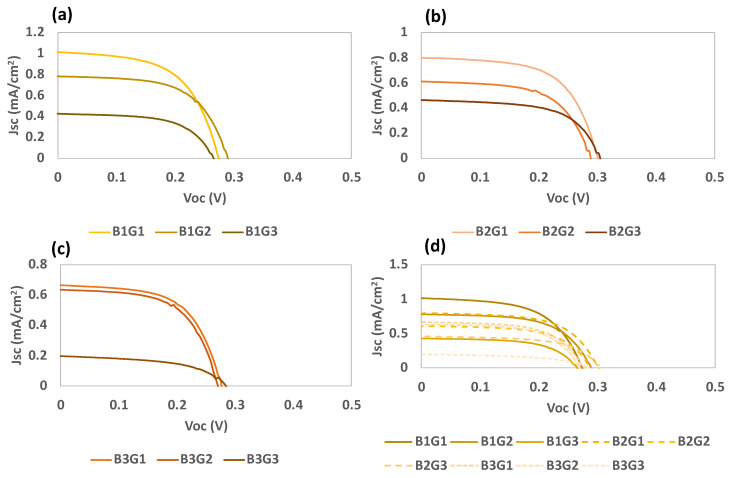
Photocurrent–voltage (J–V) curves of DSSCs based on RIO-60 as a dye replacement for the different batches, (**a**) Batch 1 (B1), (**b**) Batch 2 (B2), (**c**) Batch 3 (B3), and (**d**) all batches under simulated AM 1.5 G illumination at 100 mW/cm^2^.

**Figure 13 nanomaterials-13-01204-f013:**
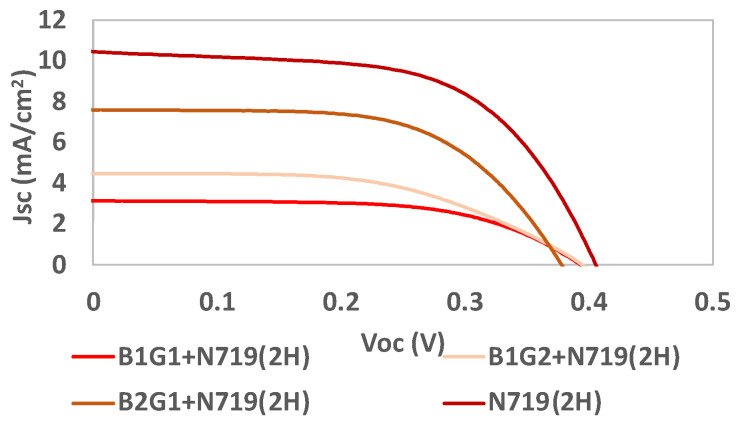
Photocurrent–voltage (J–V) curves of DSSCs based on the use of the COF RIO-60 as an interface between TiO_2_ and the reference dye N719 for the different batches B1G1, B1G2, B2G1, and N719, all under simulated AM 1.5G illumination at 100 mW/cm^2^.

**Table 1 nanomaterials-13-01204-t001:** Photovoltaic performance parameters of DSSCs based on the use of COF starting materials as dyes after adsorption for 5 h, under simulated AM 1.5 G illumination at 100 mW/cm^2^. (*The presented results correspond to the best-performing cell*).

Parameters	BBY	DAPSO	DAAQ	TFPG	N719
**FF ***	0.52 ± 0.00	0.49 ± 0.02	0.41 ± 0.08	0.60 ± 0.00	0.59 ± 0.01
**η (%) ***	0.05 ± 0.00	0.03 ± 0.01	0.04 ± 0.01	0.41 ± 0.00	3.06 ± 0.41
**V_oc_ (mV) ***	295 ± 4	299 ± 36	300 ± 36	317 ± 5	432 ± 20
**J_sc_ (mA/cm^2^) ***	0.33 ± 0.00	0.21 ± 0.03	0.30 ± 0.01	2.12 ± 0.05	11.88 ± 1.09

* all the values are presented with their respective standard deviations.

**Table 2 nanomaterials-13-01204-t002:** Photovoltaic performance parameters of DSSCs based on COFs RIO-60 as a dye replacement. The initial assessment was conducted under simulated AM 1.5 G illumination at 100 mW/cm^2^. (*The presented results correspond to the best-performing cell*).

Parameters	RIO-70	RIO-43	RIO-55	RIO-60	COF-DAAQ
**FF ***	0.43 ± 0.00	0.28 ± 0.00	0.62 ± 0.01	0.59 ± 0.00	0.62 ± 0.00
**η (%) ***	0.01 ± 0.00	0.00 ± 0.00	0.03 ± 0.00	0.18 ± 0.01	0.07 ± 0.00
**V_oc_ (mV) ***	263 ± 3	44 ± 0	339 ± 1	286 ± 1	424 ± 3
**J_sc_ (mA/cm^2^) ***	0.10 ± 0.00	0.02 ± 0.00	0.15 ± 0.00	1.04 ± 0.02	0.25 ± 0.00

* all values are presented with their respective standard deviations.

**Table 3 nanomaterials-13-01204-t003:** Photovoltaic performance parameters of DSSCs based on COFs and RIO as dye replacements using an optimized deposition step under simulated AM 1.5 G illumination at 100 mW/cm^2^. (*The presented results correspond to the best-performing cell*).

Parameters	RIO-55	RIO-60	COF-DAAQ
**FF ***	0.51 ± 0.02	0.59 ± 0.02	0.59 ± 0.01
**η (%) ***	0.05 ± 0.01	0.19 ± 0.00	0.11 ± 0.00
**V_oc_ (mV) ***	284 ± 2	329 ± 7	307 ± 2
**J_sc_ (mA/cm^2^) ***	0.37 ± 0.07	1.00 ± 0.01	0.60 ± 0.01

* all values are presented with their respective standard deviations.

**Table 4 nanomaterials-13-01204-t004:** Impregnation conditions (batch and deposition time) used for COF RIO-60.

Deposition Time	Batch 1 (½ h for RIO-60 Formation)	Batch 2 (2 h for RIO-60 Formation)	Batch 3 (24 h for RIO-60 Formation)
**30 min**	B1G1	B2G1	B3G1
**5 h**	B1G2	B2G2	B3G2
**24 h**	B1G3	B2G3	B3G3

**Table 5 nanomaterials-13-01204-t005:** Photovoltaic performance parameters of DSSCs based on the use of the COF RIO-60 as a dye replacement for the different batches under simulated AM 1.5 G illumination at 100 mW/cm^2^. (*The presented results correspond to the best performing cell*).

Parameters	B1G1	B1G2	B1G3	B2G1	B2G2	B2G3	B3G1	B3G2	B3G3
**FF ***	0.58 ± 0.00	0.60 ± 0.01	0.59 ± 0.00	0.60 ± 0.00	0.61 ± 0.01	0.61 ± 0.01	0.58 ± 0.01	0.55 ± 0.04	0.52 ± 0.02
**η (%) ***	0.13 ± 0.03	0.13 ± 0.01	0.07 ± 0.00	0.14 ± 0.00	0.10 ± 0.00	0.09 ± 0.00	0.11 ± 0.00	0.09 ± 0.01	0.03 ± 0.00
**V_oc_ (mV) ***	274 ± 1	278 ± 12	268 ± 3	301 ± 0	281 ± 8	305 ± 5	274 ± 13	262 ± 8	283 ± 17
**J_sc_(mA/cm^2^) ***	0.84 ± 0.18	0.77 ± 0.03	0.42 ± 0.00	0.79 ± 0.01	0.61 ± 0.01	0.48 ± 0.01	0.70 ± 0.05	0.63 ± 0.03	0.20 ± 0.00

* all values are presented with their respective standard deviations.

**Table 6 nanomaterials-13-01204-t006:** Photovoltaic performance parameters of DSSCs based on the use of COF RIO-60 as an interface between TiO_2_ and the reference dye N719 for the best-performance batch under simulated AM 1.5G illumination at 100 mW/cm^2^. (*The presented results correspond to the best-performing cell*).

Parameters	B1G1+N719 (2 h)	B1G2+N719 (2 h)	B2G1+N719 (2 h)	N719 (2 h)
**FF ***	0.59 ± 0.03	0.52 ± 0.06	0.61 ± 0.07	0.56 ± 0.02
**η (%) ***	0.74 ± 0.03	0.93 ± 0.13	1.74 ± 0.43	2.24 ± 0.05
**V_oc_ (mV) ***	386 ± 7	395 ± 9	378 ± 5	417 ± 5
**J_sc_(mA/cm^2^) ***	3.26 ± 0.08	4.49 ± 0.28	7.61 ± 1.39	9.66 ± 0.37

* all values are presented with their respective standard deviations.

## Data Availability

Not applicable.

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
