# Peer review of "Application of Covalent Organic Frameworks (COFs) as Dyes and Additives for Dye-Sensitized Solar Cells (DSSCs)"

_nanomaterials, 2023, doi:10.3390/nano13071204_

Round 1
Reviewer 1 Report
1. Author propsed 5 COFs for DSSC application, but only RIO-66 was characterized by UV-vis, FTIR and BET. I am quite interested in other 4 COFs.
2. Based on your chemical synthesis, how did you characterize the desired chemical structures as shown in Fig. 2 ?
3. Too much J-V curves plots of DSSCs. It is hard to understand what you want to elicidate. Please make these tables more concisely.
4. Some references regarding the photovoltaics could give readers basic undertstanding, such as organic photovoltaics. I recommed these two refs. Materials Today 2012, 15 (12), 554-562 and Journal of Materials Chemistry A 2017, 5, 24051-24075 could be cited in the introduction.
Author Response
Reviewer 1 (R1) considered that the Author proposed 5 COFs for DSSC application, but only RIO-66 was characterized by UV-vis, FTIR and BET. I am quite interested in other 4 COFs.
Authors: Thank you for your comment. It is important to clarify that the COF RIO-60 is a new material and for that reason a more detailed characterization by UV-vis, FTIR and BET including textural analysis. The other 4 COFs were previously reported but the characterization were included in the ESI.
R1: Based on your chemical synthesis, how did you characterize the desired chemical structures as shown in Fig. 2?
Authors: All materials were characterized by N2 adsorption/desorption isotherms (BET method for surface area and NLDFT for pore size distribution), FTIR and solid-state 13C NMR (to observe and confirm the new bonds in the arrangement), and PXRD. All of these characterizations were included in the ESI.
R1: Too much J-V curves plots of DSSCs. It is hard to understand what you want to elicidate. Please make these tables more concisely.
Authors: Thank you for your comment. The inclusion of more J-V curves plots of DSSCs allowed us to elucidate if the starting materials interfere with the measurements. Then, the best COF for the application in DSSCs defined the best adsorption conditions. We tried to make the tables more concise.
R1: Some references regarding the photovoltaics could give readers basic undertstanding, such as organic photovoltaics. I recommed these two refs. Materials Today 2012, 15 (12), 554-562 and Journal of Materials Chemistry A 2017, 5, 24051-24075 could be cited in the introduction.
Authors: Thank you for your comment. The references were included in the introduction.
Best regards
Hugo Cruz
Reviewer 2 Report
Dear Authors
The manuscript is focused on the five different Covalent Organic Frameworks (COFs) synthesized and applied in Dye-Sensitized Solar Cells (DSSCs) as dye and additive (as an interface between the dye and photoanode).
The following suggestion and comments should be taken:
1. The overall English needs to be improved. Please seek guidance from a native English speaker if possible ("the" "a", commas, plural form and others could be corrected).
2. The introduction section needs enhancement few sentences about other hybrid carbon materials for DSSC. applications. Please cite:
(1) Electronics 2023, 12(3), 570; https://doi.org/10.3390/electronics12030570
(2) Materials 2019, 12(20), 3354; https://doi.org/10.3390/ma12203354
(3) Micromachines 2023, 14(2), 394; https://doi.org/10.3390/mi14020394
3. Please better explain the novelty of your work.
4. Figure 2 please correct this image for better quality.
5. Figure 4. Please correct this image for better quality (the inscriptions).
6. Could the authors include the standard deviation of the used methods?
7. Are authors changing some parameters to improve efficiency? What is the main impact factor?
8. Figure 10. Please correct this image for better quality (the inscriptions).
9. Why author choose these systems for the study? Please explain.
10. Authors are suggested to describe some future plans in conclusions.
Author Response
Reviewer 2 (R2) considered that “The manuscript is focused on the five different Covalent Organic Frameworks (COFs) synthesized and applied in Dye-Sensitized Solar Cells (DSSCs) as dye and additive (as an interface between the dye and photoanode).”
R2: The overall English needs to be improved. Please seek guidance from a native English speaker if possible ("the" "a", commas, plural form and others could be corrected).
Authors: Thank you for your comment. We tried to improve the overall English of the manuscript.
R2: The introduction section needs enhancement few sentences about other hybrid carbon materials for DSSC applications. Please cite: (1) Electronics 2023, 12(3), 570; https://doi.org/10.3390/electronics12030570; (2) Materials 2019, 12(20), 3354; https://doi.org/10.3390/ma12203354; (3) Micromachines 2023, 14(2), 394; https://doi.org/10.3390/mi14020394.
Authors: The references were included in the manuscript.
R2: Please better explain the novelty of your work.
Authors: The novelty of this work is based on the application of new family of COFs called Reticular Innovative Organic frameworks (RIO) in Dye Sensitizer Solar Cells (DSSCs). These nanomaterials are composed by ionic organic dyes (e.g. thionin acetate, Bismarck brown Y, and pararosaniline hydrochloride) as well as antibiotics (e.g. dapsone, a commercial electroactive compound) building blocks and to test as dyes and solid additives in DSSCs. In this context, RIO-60 is a new material possessing interesting properties for energy storage and production.
R2: Figure 2 please correct this image for better quality.
Authors: The figure 2 was corrected as indicated by reviewer.
R2: Figure 4. Please correct this image for better quality (the inscriptions).
Authors: The figure 4 was corrected as indicated by reviewer.
R2: Could the authors include the standard deviation of the used methods?
Authors: The standard deviation of the used methods are included in the tables as suggested.
R2: Are authors changing some parameters to improve efficiency? What is the main impact factor?
Authors: In these preliminary studies, the absorption time as well as the use of COF as an additive to reference N719 were optimized in order to improve the efficiency. One of the objectives for the future will be to design a COF to improve the current density (JSC). The obtained results can facilitate the development of new COFs for future work.
R2: Figure 10. Please correct this image for better quality (the inscriptions).
Authors: The figure 10 was corrected as indicated by reviewer.
R2: Why author choose these systems for the study? Please explain.
Authors: The selection of these systems was based on the expertise to develop COFs and the potential of their use as materials for DSSCs. In this context, the prepared COFs showed higher stability (covalent bonds forming a robust arrangement), suitable porosity and chemical structure (allowing the guest several molecules and ions, interacting with the lattice and creating the desired processes).
It is important to note that cheap, abundant and commercial available building blocks based on ionic dyes and antibiotics were selected. The main idea will be to use novel porous nanomaterials in order to obtain a better performance of the solar cell, leading to a long-lasting solar cell.
R2: Authors are suggested to describe some future plans in conclusions.
Authors: Thank you for your suggestion. Some plans were included in the conclusion.
Best regards
Hugo Cruz
Reviewer 3 Report
Manuscript ID: nanomaterials-2284159. Title: Application of Covalent Organic Frameworks (COFs) as Dye and additives for Dye-Sensitized Solar Cells (DSSCs). In this paper the author five different Covalent Organic Frameworks (COFs) were synthesized and applied in Dye-Sensitized Solar Cells (DSSCs) as dye and additive (as interface between the dye and photoanode). The author claiming that this first approach using RIO´s family opens good perspectives for application in DSSCs in the role of dye or photoanode dye enhancer helping to improve the cell lifetime. Their presentation on review writing was not so excited. I am recommending this review because of only for few articles reported in solar PV system, Covalent Organic Frameworks (COFs) as Dye and additives for Dye-Sensitized Solar Cells (DSSCs). It’s an interesting topic. Therefore, I recommend publication only after minor revisions.
1)The abstract of the review article is not so exciting; the author needs to rewrite the abstract (needs to include the motivation and necessity of this research).
2) In Figure 2 and Figure 6, the author needs to include same size of all the compounds. Some of the compounds are shown as smaller sizes and some of the compounds are shown as bigger sizes.
3) In Figure 6, the author included yellow compounds in one glass vial and another (like a beaker), which needs to indicate what are the compounds.
4) figure 7, which figure (a) and (b), and (c)…. needs to indicate correctly.
5) All the figures need check and correct.
Author Response
Reviewer 3 considered that five different Covalent Organic Frameworks (COFs) were synthesized and applied in Dye-Sensitized Solar Cells (DSSCs) as dye and additive (as interface between the dye and photoanode). The author claiming that this first approach using RIO´s family opens good perspectives for application in DSSCs in the role of dye or photoanode dye enhancer helping to improve the cell lifetime. Their presentation on review writing was not so excited. I am recommending this review because of only for few articles reported in solar PV system, Covalent Organic Frameworks (COFs) as Dye and additives for Dye-Sensitized Solar Cells (DSSCs). It’s an interesting topic. Therefore, I recommend publication only after minor revisions.
R3: The abstract of the review article is not so exciting; the author needs to rewrite the abstract (needs to include the motivation and necessity of this research).
Authors: Thank you for your comment. The abstract of article was changed according to the reviewer comments.
R3: In Figure 2 and Figure 6, the author needs to include same size of all the compounds. Some of the compounds are shown as smaller sizes and some of the compounds are shown as bigger sizes.
Authors: The figures 2 and 6 were changed according to the comment.
R3: In Figure 6, the author included yellow compounds in one glass vial and another (like a beaker), which needs to indicate what are the compounds.
Authors: The figure 6 was changed according to the comment.
R3: Figure 7, which figure (a) and (b), and (c)…. needs to indicate correctly.
Authors: The figure 7 was corrected.
R3: All the figures need check and correct.
Authors: All figures were checked and corrected as indicated.
Best regards,
Hugo Cruz
Round 2
Reviewer 1 Report
Author answered all the question and provided valueable good data. However, when I read the version 2 manuscript. I found Figure 7 (new data) was added. No any explanation paragraph for figure 7. Also I suggest author need to write "Figure xx" in the paragraph of your manuscript, helping reader to connect your paragraph and figure. This is the basic minimun requirement for academic writing.
Author Response
Comments and Suggestions for Authors
Author answered all the question and provided valueable good data. However, when I read the version 2 manuscript. I found Figure 7 (new data) was added. No any explanation paragraph for figure 7. Also I suggest author need to write "Figure xx" in the paragraph of your manuscript, helping reader to connect your paragraph and figure. This is the basic minimun requirement for academic writing.
Authors: Thank you for your comment. According to the reviewer suggestions an explanation paragraph for figure 7was added.
The paragraph included in the document is:
In Figure 7 is presented the physic-chemical Characterization of RIO-60 a) FTIR, b) solid state 13C NMR, c) PXRD, and d) BET method for surface area N2 adsorption–desorption isotherm and pore size distribution. Figure 7a) FTIR spectroscopy characterization is frequently used to determine the compositional structure the materials. The spectrum obtained in Figure 7a) for the RIO-60 showed the characteristic signal for the RIO-60. A broad band at 3480 cm-1 are attributed to the -C-N while signal at 1628 cm-1 related to -C=O. The bands at 1330 and 1180 cm-1 that are associated with -SO2 asymmetric and -SO2 symmetric stretch. Figure b) solid-state 13C CP-MAS NMR spectrum characterization of RIO-60
is also frequently used complete the determination the compositional structure of
COFs. The spectrum obtained for RIO-60 carbon are presented in Figure 7b) I different color associated to the carbon assigned: in pink dot (184.88 ppm),in green dot (142.57 ppm), blue dot (138.04 ppm), red dot (130.03 ppm), purple dot (119.63 ppm) and yellow dot 108.08 ppm, respectively.
Characterization by PXRD using powder XRD see Figure 7c) the diffractogram acquired shows a low crystallinity profile, with a broad peak at ~25o.
In figure 7d) is presented the characterization by Brunauer–Emmet–Teller (BET) specific surface area obtained by N2adsorption–desorption isotherm at 77 K RIO-60. The pore size distribution (see the inset of Figure 7d), calculated by NLDFT model from desorption branch N2 at 77 K on carbon cylindrical/slit pores, showed pore size: 13 AÌŠ.
The paragraph is highlight in green.
The Authors would like to thank you for your comment.
Best Regards
Reviewer 2 Report
The authors have addressed all comments and the manuscript can be published as is.
Author Response
The Authors would like to thank you for your comment.
Best Regards